# VARX Granger analysis: Models for neuroscience, physiology, sociology and econometrics

**Lucas C. Parra**[1]*, **Aimar Silvan**[1], **Maximilian Nentwich**[2], **Jens Madsen**[1], **Vera E. Parra**[3], **Behtash Babadi**[4]*

**1** Department of Biomedical Engineering, City College of New York, New York, NY, United States of America, **2** Institute of Bioelectronic Medicine, Northwell Health Feinstein Institutes for Medical Research, Manhasset, NY, United States of America, **3** Department of Sociology, University of California Berkeley, Berkeley, CA, United States of America, **4** Department of Electrical and Computer Engineering, University of Maryland, College Park, MD, United States of America

* parra@ccny.cuny.edu (LCP); behtash@umd.edu (BB)

## Abstract

Complex systems, such as in brains, markets, and societies, exhibit internal dynamics influenced by external factors. Disentangling delayed external effects from internal dynamics within these systems is often difficult. We propose using a Vector Autoregressive model with eXogenous input (VARX) to capture delayed interactions between internal and external variables. Whereas this model aligns with Granger's statistical formalism for testing "causal relations", the connection between the two is not widely understood. Here, we bridge this gap by providing fundamental equations, user-friendly code, and demonstrations using simulated and real-world data from neuroscience, physiology, sociology, and economics. Our examples illustrate how the model avoids spurious correlation by factoring out external influences from internal dynamics, leading to more parsimonious explanations of these systems. For instance, in neural recordings we find that prolonged response of the brain can be explained as a short exogenous effect, followed by prolonged internal recurrent activity. In recordings of human physiology, we find that the model recovers established effects such as eye movements affecting pupil size and a bidirectional interaction of respiration and heart rate. We also provide methods for enhancing model efficiency, such as L2 regularization for limited data and basis functions to cope with extended delays. Additionally, we analyze model performance under various scenarios where model assumptions are violated. MATLAB, Python, and R code are provided for easy adoption: https://github.com/lcparra/varx.

## 1 Introduction

Analyzing signals generated by real-world dynamical systems such as neural activity in the brain, physiological signals in the body, or trends in society and the economy is a key component of scientific discovery. These systems involve endogenous variables, which are internal variables that develop and interact with each other over time. Additionally, these systems are

**Data Availability Statement:** https://github.com/lcparra/varx.

**Funding:** This work was supported by the National Institutes of Health (NIMH P50 MH109429 to LCP,

ASO and JM) and by the National Science
Foundation (DRL-2201835 to LCP and JM).

**Competing interests:** The authors have declared
that no competing interests exist.

influenced by exogenous variables, which are external factors that serve as drivers of the endogenous dynamics (for instance, a visual stimulus to the brain, or fiscal stimulus to the economy). It is often not clear how to separate the external drive from the internal dynamics.

A standard modeling approach to capture effects between dynamic variables is to determine if one variable can be predicted from another. To determine whether temporal predictions capture real statistical effects, Clive Granger proposed a method to compute statistical significance in his seminal work on causality [1]. He asked whether the quality of the prediction is significantly improved when a variable is added to the model or not. If such an improvement is observed, we say that $x$ has an "effect" on $y$. Granger referred to this as a "causal relation". The basic idea had been suggested earlier by Wiener [2], and it is sometimes referred to as "Wiener-Granger Causality" [3]. In this work, we avoid calling an effect "causal", due to several well-known limitations of this interpretation, which we will discuss.

Conventionally, one focuses on linear prediction in parametric multivariate models, which often captures the dominant relationships between dynamic variables. When considering time delays, this translates to finding a linear filter that best predicts the next time point based on the preceding signal. When predicting a variable from its own past, this filter is referred to as an auto-regressive (AR) model. In scenarios with multiple endogenous variables, a vectorial auto-regressive (VAR) model is employed, characterized by multiple filters between all the variables. Finally, when some variables represent exogenous inputs, the corresponding model is known as a VARX model [4].

In practice, optimal linear filters are estimated using ordinary least squares. Then, the logarithmic ratio of the two prediction error variances, in presence and absence of $x(t)$ and its past, is taken as the test statistic and its statistical significance is assessed based on the corresponding asymptotic distributions [5–7]. While this procedure is relatively simple to perform, it faces a key challenge: in order to obtain reliable parameter estimates via least squares, a typically long observation window is required. In datasets with short duration, the foregoing models typically over-fit the observed data, resulting in unreliable parameter estimates [8, 9].

This challenge has been addressed in the context of regularized estimation [10–19], in which the least squares objective function is augmented with a penalty term that enforces additional restrictions on the parameters, such as smoothness, sparsity, and low-rank structures. While some of these sophisticated regularization schemes, such as L1-regularization, or the LASSO [20, 21], smoothly clipped absolute deviation [22, 23], Elastic-Net [24], and their variants have particularly proven useful in VAR estimation [11–13, 17, 18], regularization with the $\ell_2$-norm [10, 25], also known as "ridge regression", is arguably the most widespread in practice, due to the simplicity of the resulting parameter estimator.

It is worth noting that in parallel to the aforementioned parametric models, some existing nonparametric methods bypass VAR estimation by instead employing techniques such as spectral matrix factorization [26] or multivariate embedding [27]. In terms of statistical testing of the effects, partial correlation-based nonparametric methods that employ conditional independence tests have also been suggested [28–30], which do not require time-series modeling assumptions.

The statistical formalism developed by Granger can be applied naturally to determine the significance of the effects in VAR models [31]. Granger analysis for VAR models has been useful to neuroscientists, economists, and sociologists because it allows one to quantify the strength and direction of effects in interactive dynamical systems, such as brains [3, 8], markets [32], and societies [33]. However, researchers typically ignore exogenous variables during Granger analysis, due to a lack of tools to do so. While Granger and later Geweke allow for exogenous variables [34], these are only included to remove potential instantaneous confounds.

In this work, we introduce *VARX Granger Analysis*, with the following novel contributions. First, we treat exogenous variables in their own right by applying the Granger formalism to VARX models. By doing this, we are capturing the lagged effects of the exogenous inputs, and separating that from the internal dynamics. To enable this, we present the basic equations required for the VARX model to estimate parameters, effect size, and statistical significance based on Deviance. Then, we demonstrate the validity of this approach using simulated data. We further explore methods, such as L2 regularization and the usage of basis functions in parametric modeling, to handle high-dimensional datasets and obtain longer prediction filters. These methods effectively reduce the number of parameters. We present in a Supplement the first derivation of de-biased estimates of the test statistic for L2 regularization. We will also illustrate instances where interpreting the Granger formalism as "causal" can be misleading, such as in cases involving missing variables or colliders. Subsequently, we showcase examples that apply this formalism to neural signals, highlighting the key differences between VARX models and "temporal response functions" [35] commonly used in neuroscience. We will also present examples using physiological, sociological, and economic data, treated here for the first time with the VARX formalism. Finally, we conclude with a discussion on the specifics of code implementation and some caveats regarding the interpretation of model results.

## 2 Methods

A way to understand the VARX model is to imagine pellets dropping into a pond. The pellets are like an external input (exogenous), but the ripples they create are governed by the water's own internal dynamics (endogenous). These ripples can also be influenced by unpredictable wind gusts. Our goal is to distinguish between these external and internal factors by analyzing what we can observe (the pellets and the pattern of the surface of the water) while minimizing the influence of the unseen wind.

### 2.1 VARX model

More formally, consider the vectorial "input" signal $\mathbf{x}(t)$ and the vectorial "output" signal $\mathbf{y}(t)$ of dimensions $d_x$ and $d_y$ respectively, with both assumed to be observable (lower-case bold characters represent vectors). In the case of brain activity, the input may be multiple features of a continuous natural stimulus, say, luminance and sound volume of a movie. The output could be neural activity recorded at multiple locations in the brain (see Section 3.3). In the case of macroeconomic variables, the input could be government spending and the endogenous variables could be various indicators of economic activity of a nation (see Section 3.6). The simplest model we can envision is one in which the current signal $\mathbf{y}(t)$ can be predicted linearly from the input $\mathbf{x}(t)$ and also linearly from the preceding output $\mathbf{y}(t - 1)$ (Fig 1).

$$\mathbf{y}(t) = \mathbf{A} * \mathbf{y}(t - 1) + \mathbf{B} * \mathbf{x}(t) + \mathbf{e}(t). \tag{1}$$

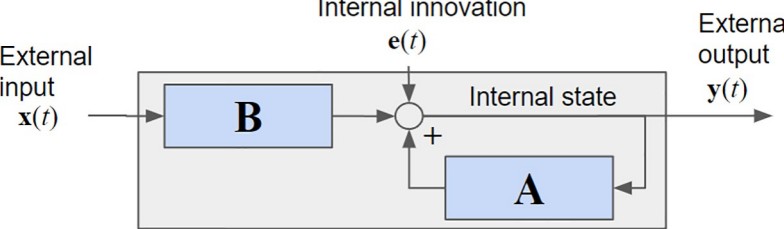

**Fig 1. VARX model: The gray box represents the overall system response $\mathcal{H}$.**

Here, $\mathbf{A}$ and $\mathbf{B}$ are filter matrices of dimensions $[d_y, d_y]$ and $[d_y, d_x]$, with filters of length $n_a$ and $n_b$, respectively. The additive term $\mathbf{e}(t)$ represents an unobserved "innovation" that introduces an error in the prediction. In the theory of linear systems this is called innovation because it injects novelty into the recurrent dynamic. We refer to $\mathbf{y}(t)$ as endogenous variables, as they are influenced by one another including their own history (through the diagonal terms in $\mathbf{A}$, and to $\mathbf{a}(t)$ as exogenous variable as they are fixed and not influenced by the endogenous variables.

In Eq (1) we have used a compact formulation of a multi-input multi-output convolution, which for the auto-regressive filters $\mathbf{A}$ and moving average filters $\mathbf{B}$ reads:

$$\mathbf{A} * \mathbf{y}(t-1) \quad = \quad \sum_{l=1}^{n_a} \mathbf{A}(l) \cdot \mathbf{y}(t-l), \tag{2}$$

$$\mathbf{B} * \mathbf{x}(t) \quad = \quad \sum_{l=0}^{n_b-1} \mathbf{B}(l) \cdot \mathbf{x}(t-l). \tag{3}$$

## 2.2 Total system response

Note that the total response of the dynamical system (the impulse response) can be written in the $Z$ or Fourier domains simply as

$$\mathcal{H} = \mathcal{Y}\mathcal{X}^{-1} = (\mathcal{I} - \mathcal{A})^{-1}\mathcal{B}. \tag{4}$$

In this view, what we are proposing is to model the total system response as a combination of a Moving Average (MA) filtering $\mathbf{B}$ and Auto-Regressive (AR) filtering $(\mathbf{I} - \mathbf{A})^{-1}$. In the time domain, the total system response $\mathbf{H}(t)$ can simply be computed by passing impulses in each input variable through the system, while setting the error/innovation to zero, $\mathbf{e}(t) = 0$. The alternative is to model the total system response as a single MA filter. This is the approach taken in mTRF [35] and Neuro-Current Response Function (NCRF) [36] frameworks. We will show (in Fig 5) that the total response estimated either as VARX or MA models are practically the same. However, as we see in Eq (4), the VARX model factors the total response into an exogenous and an endogenous dynamic, with parameters $\mathbf{B}$ and $\mathbf{A}$, respectively.

## 2.3 System identification

Given observed $\mathbf{x}(t)$ and $\mathbf{y}(t)$ one can estimate the parameters $\mathbf{A}$ and $\mathbf{B}$ by minimizing the mean of square error:

$$\boldsymbol{\sigma}^2 = \frac{1}{T}\sum_{t=1}^{T} \mathbf{e}^2(t). \tag{5}$$

This identification criterion is equivalent to Maximum Likelihood estimation, if one assumes that the innovation $\mathbf{e}(t)$ is normally distributed and uncorrelated (spherical and white). For zero-mean signals this is also the variance, hence the conventional symbol $\boldsymbol{\sigma}^2$, which is vectorial here as it is computed and minimized for each dimension in $\mathbf{y}$ individually. For the VARX model, the linear predictors $\mathbf{y}(t-1)$ and $\mathbf{x}(t)$ are observable (this will not be true for the Output Error model discussed in Section 2.8), and parameter estimation results in a simple linear least-squares problem with a well-established closed-form solution. Eq (1) can be rewritten as

$$\mathbf{Y} = \mathbf{X} \cdot \mathbf{H} + \mathbf{E}, \tag{6}$$

where **X** is a block-Toeplitz matrix of the predictors, including $\mathbf{y}(t-1)$ and $\mathbf{x}(t)$. This matrix has dimensions $[T, N]$, where $N$ is the total number of free parameters for each predicted dimension in $y(t)$. In this case $N = d_y n_a + d_x n_b$. **Y** is the output signal $\mathbf{y}(t)$ arranged as a matrix of dimensions $[T, d_y]$, and $\mathbf{H} = [\mathbf{A}, \mathbf{B}]^\top$ is a matrix of dimensions $[d_y, N]$ combining the AR and MA filters. The least-squares estimate is then simply:

$$\hat{\mathbf{H}} = \mathbf{R}_{xx}^{-1} \mathbf{R}_{xy}. \tag{7}$$

Matrices $\mathbf{R}_{xx}$ and $\mathbf{R}_{xy}$ are block-Toeplitz capturing cross-correlations:

$$\mathbf{R}_{xx} = \mathbf{X}^\top \cdot \mathbf{X}, \tag{8}$$

$$\mathbf{R}_{xy} = \mathbf{X}^\top \cdot \mathbf{Y}. \tag{9}$$

The estimated output is:

$$\hat{\mathbf{Y}} = \mathbf{X} * \hat{\mathbf{H}}. \tag{10}$$

The residual errors of this model prediction for each output channel are the diagonal elements of the correlation matrix for the errors:

$$\mathbf{R}_{ee} = (\mathbf{Y} - \hat{\mathbf{Y}})^\top \cdot (\mathbf{Y} - \hat{\mathbf{Y}}) = \hat{\mathbf{H}}^\top \cdot \mathbf{R}_{xx} \cdot \hat{\mathbf{H}} - 2\hat{\mathbf{H}}^\top \cdot \mathbf{R}_{xy} + \mathbf{R}_{yy}, \tag{11}$$

$$\boldsymbol{\sigma}^2 = \frac{1}{T}\mathbf{diag}(\mathbf{R}_{ee}). \tag{12}$$

In Section 2.7 we discuss how these expressions change when we employ L2 regularization to mitigate overfitting. In addition, we will extend the approach to include basis functions to represent filters **B**, which reduce the number of free parameters, again with the goal of reducing overfitting.

Note that Eqs (2)–(10) are identical to modeling the total system response as a single multivariate MA filter, whereby matrix **X** only contains the input $\mathbf{x}(t)$, with $N = d_x n_b$, and the impulse response $\mathbf{H} = \mathbf{B}$. In the neuroscience literature, this MA model is referred to as a "multivariate Temporal Response Function" (mTRF).

## 2.4 Granger formalism

To establish whether any of the channels in filters **A** or **B** significantly improve predictions, i.e. have an "effect", one can use a likelihood-ratio test [37]. In this formalism, one uses Deviance as test statistics to quantify the contribution of a given predictor in **X** for each output in **Y**. The approach consists of estimating filter parameters **H** with all predictors included in **X**, which is referred to as the "full" model, and then again with one of the predictors removed, which is referred to as the "reduced" model. We compute the resulting square error $\hat{\boldsymbol{\sigma}}_f^2$ and $\hat{\boldsymbol{\sigma}}_r^2$, for the full and reduced models, and obtain the *deviance* between the two models as test statistics (there is one Deviance value for each dimension in **y**):

$$\mathscr{D} := T \log\left(\hat{\boldsymbol{\sigma}}_r^2 / \hat{\boldsymbol{\sigma}}_f^2\right), \tag{13}$$

where the division of the two variance vectors and the log operator are interpreted element-wise. For normal, independently, and identically distributed error, the vector $\mathscr{D}$ contains the log-likelihood ratios (times a factor of 2), with each element following a chi-square distribution [7]. Notice that the test statistics vector $\mathscr{D}$ is formed by computing the log-likelihood ratio for each output dimension, and for each predictor dimension that is removed in the reduced

model. Thus, one can estimate the statistical significance of each channel in **A** and **B** by computing the full model once, and then removing each predictor variable individually from the full model. The statistical significance for a non-zero contribution from a particular predictor to a particular output is then given by an element of the "p-value" vector computed with the corresponding Deviance vector:

$$\mathbf{p} = \mathbf{1} - F(\mathcal{D}, n), \tag{14}$$

Here $F$ is the cumulative distribution function for the chi-square distribution and $n$ is the number of parameters that were removed in the reduced model, i.e $n_a$ or $n_b$ depending on whether an element of $\mathbf{y}(t-1)$ or $\mathbf{x}(t)$ was removed. The operation of $F$ on a vector is interpreted element-wise.

## 2.5 Effect size

Note that Deviance increases linearly with $T$, that is, the statistical evidence increases with the length of the signals, and thus cannot be used as effect size. A traditional definition of effect size in the context of reduced and full linear models is the coefficient of determination, or generalized R-square [38]:

$$R^2 := \mathbf{1} - \exp(-\mathcal{D}/T), \tag{15}$$

where the exponential of a vector is interpreted element-wise.

## 2.6 Debiased deviance for L2 regularization

To avoid overtraining with small sample sizes, i.e. where $T$ is not much larger than $N$, we decided to use an L2 penalty, with Tikhonov regularization. The advantage over other forms of regularization, such as L1 [19, 39] or a state space model [40] is computational efficiency thanks to the closed-form solution:

$$\hat{\mathbf{H}} = (\mathbf{R}_{xx} + \gamma \mathbf{\Gamma})^{-1} \mathbf{R}_{xy}, \tag{16}$$

where we selected $\mathbf{\Gamma} = \text{diag}(\mathbf{R}_{xx})$ so that all variables are regularized equally regardless of their scale, the choice of $\gamma$ is discussed in the results section. This regularization introduces (purposefully) a bias in the estimate, and the deviance estimate has to be corrected to account for this bias [41]. The derivation for the term that corrects the log-likelihood in case of L2 regularization is available in the S3 File:

$$\mathbf{b} = \frac{1}{2} \text{diag}(\mathbf{R}_{xe}^{\top} \cdot \mathbf{R}_{xx}^{-1} \cdot \mathbf{R}_{xe}) / \text{diag}(\mathbf{R}_{ee}), \tag{17}$$

where the division between the two diagonal vectors is element-wise and $\mathbf{R}_{xe} = \mathbf{R}_{xy} - \mathbf{R}_{xx} \cdot \hat{\mathbf{H}}$. This bias term has to be computed for the full and reduced models, giving $\mathbf{b}_f$ and $\mathbf{b}_r$ respectively. The corresponding de-biased deviance is then:

$$\mathcal{D}^{de-biased} = T \log(\hat{\boldsymbol{\sigma}}_r^2 / \hat{\boldsymbol{\sigma}}_f^2) - \mathbf{b}_r + \mathbf{b}_f, \tag{18}$$

and can be used to compute the p-values as before. We have found empirically that we obtain a better (conservative) estimate of p-values if we use $T' = T - N$ instead of $T$ in this calculation of the de-biased deviance. $T'$ represents the effective degrees of freedom of the full model and converges to $T$ in the asymptotic limit for which the de-biased deviance formula was derived. The choice of the regularization factor $\gamma$, and its relationship to $T$ are discussed in the S1 File.

## 2.7 Basis functions for the moving average filters

The filter length (number of parameters) used in AR filters is typically kept relatively short, to avoid over-fitting, reduce the odds of instability in the recursion, and because even a single delay can already represent an infinite impulse response. This is not the case for MA filters, where longer responses have to be modeled explicitly, which can result in a relatively large number of parameters, with a risk of over-fitting. We have found empirically that the corrections we introduced in the Deviance estimate for short signals (Eq 18) do not work well when the filter lengths for **A** and **B** are very different. A solution to both these problems (imbalance in number of parameters and filter length) is to use basis functions for the **B** filters, following an approach used previously for TRFs [42–44]. In this formalism, we have:

$$\mathbf{B} = \underline{\mathbf{B}} \circ \mathbf{W}, \tag{19}$$

where the inner product $\circ$ is along the lag-axis of the filter matrix **B**, and the goal now is to find the optimal $\underline{\mathbf{B}}$. The matrix **W** has dimensions $[n_b, \underline{n}]$ so that the number of parameters per filter is reduced from $n_b$ to $\underline{n}$. The linear least-squares problem remains unchanged with the closed-form solutions using now $\underline{\mathbf{X}} = \mathbf{W}\mathbf{X}$. In the equations above this can be implemented by replacing $\mathbf{R}_{xx}$ and $\mathbf{R}_{xy}$ with:

$$\mathbf{R}_{\underline{xx}} = \mathbf{W} \circ \mathbf{R}_{xx} \circ \mathbf{W}^{\top}, \tag{20}$$

$$\mathbf{R}_{\underline{x}y} = \mathbf{W} \circ \mathbf{R}_{xy}. \tag{21}$$

Note that the new $\mathbf{R}_{\underline{xx}}$ and $\mathbf{R}_{\underline{x}y}$ are no longer Toeplitz matrices. The Granger formalism applies without change.

Here we implemented Gaussian basis functions. With this, we are not only reducing the number of parameters, i.e. regularizing the solutions, but also selecting among a set of smooth filters **B**. In S2 File we validate the parameter and p-value estimation.

## 2.8 Equation error versus output error model

The VARX model is also called the "equation error" model [4] because the error breaks the equality of the MA and AR terms. The equation error model assumes that $\mathbf{y}(t)$ is directly observable. It is different from an "output error" model where the recursion has no error, but the recursive signal $\mathbf{z}(t)$ is hidden and only observed with additive noise (see Fig 2):

$$\begin{aligned} \mathbf{z}(t) &= \mathbf{A} * \mathbf{z}(t-1) + \mathbf{B} * \mathbf{x}(t), \\ \mathbf{y}(t) &= \mathbf{z}(t) + \mathbf{e}(t). \end{aligned} \tag{22}$$

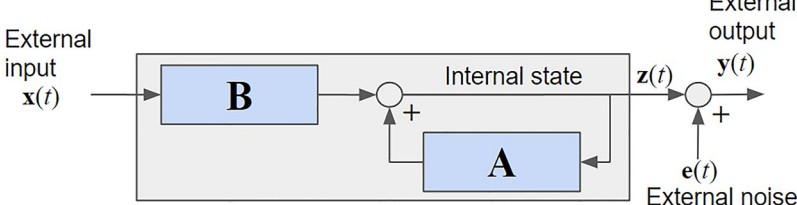

**Fig 2. Output error model: Here the dynamical variables y($t$) are not observable.** The gray box represents the overall system response $\mathcal{H}$.

For the output error model, the square error is not a quadratic function of the parameters **A** and **B**. Therefore, there is no closed-form solution to the system identification problem, as we had for the equation error model. A few different iterative optimization approaches have been proposed, such as an expectation maximization (EM) algorithm [45], gradient back-propagation through time [46, 47], or "pseudo regression" [4]. The pros and cons of the equation-error versus the output-error models are elaborated in the Discussion section.

## 3 Results

We will start with a few examples on simulated data to demonstrate the validity of the approach (Section 3.1 and 3.2). We then analyze real-world data from neuroscience, physiology, sociology and macroeconomics (Section 3.3–3.6). These examples are not meant as in-depth analyses in these diverse disciplines, but instead as demonstrations of how the VARX Granger analysis can be used in principle.

Details on all results provided next can be found in the accompanying MATLAB code repository https://github.com/lcparra/varx. Code is also provided in Python and R.

### 3.1 Test of model estimation on known model

To validate the estimation algorithm and code, we simulated a simple VARX model with two outputs and one input ($d_y = 2$, $d_x = 1$). The algorithm correctly recovers the AR and MA parameters (Fig 3)). VARX model estimation is available as part of the econometric toolbox in MATLAB but is limited to instantaneous input $n_b = 1$, i.e. no filtering of the input. When limiting the simulation to this case, the algorithms obtain similar results. Small variations are expected based on how the initial boundary conditions are handled and numerical differences. In our implementation, we omit from the estimate all samples that do not have a valid history. The code handles missing values (NaN) in the same way.

### 3.2 Validation of p-values

To validate the accuracy of the $p$-value estimation, we simulated a VARX model with all channels assigned random non-zero values, except for one channel in matrix **A** and one in matrix **B**, which were set to zero. We did this with a small and large simulated dataset, generated with

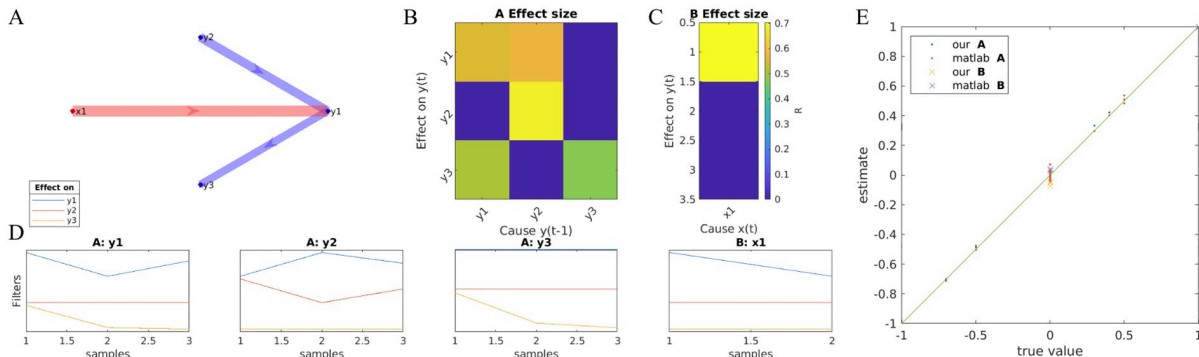

**Fig 3. Comparison of estimated parameters to true parameters in a simple toy example.** Here $d_y = 3$, $d_x = 1$, $n_a = 3$, $n_b = 1$. A: Graph shows the effect sizes $R$ indicating the structure and direction of effects (red for exogenous effect **B**, and blue for endogenous effects **A**). B: Effect sizes $R$ now shown as connectivity matrices. C: Estimated filters **A** and **B**. D: comparison of true and estimated parameters, and comparison for results from MATLAB Econometric Toolbox (we used $n_b = 1$ to satisfy the limitation of this toolbox). Signals were simulated for $T = 1000$ time steps with $x(t) \sim \mathcal{N}(0, 1)$ i.i.d., $\mathbf{e}(t)$. We used no L2 regularization and set $n_a = 3$, $n_b = 1$ for the estimation).

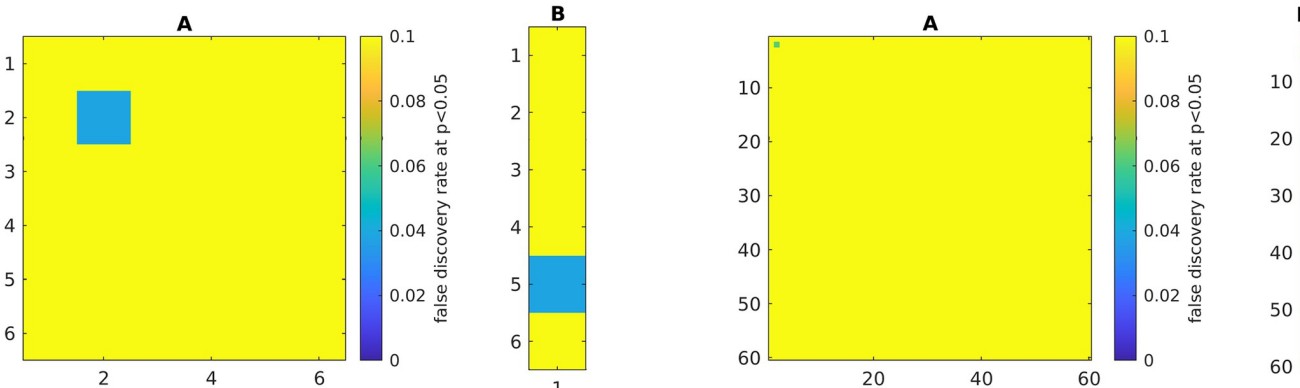

**Fig 4. Numerical validation of *p*-values in both a smaller and a larger model (*d_y* = 6 and *d_y* = 60).** Significance is set at $p < 0.05$, so we expect a false discovery rate of 0.05. Simulation here used with $d_x = 1$, $n_a = 2$, $n_b = 2$, $T = 1000$. Filter coefficients for **B** were selected at random from unit variance normal, and **A** values were set to be ±0.05 with sign selected at random (this insured stable recursion in practice). Only two channels are set to zero $A(:, 2, 2) = 0$, $B(:, 5, 1) = 0$. For these two, the false discovery is correctly estimated at approximately 0.05 (green).

normal i.i.d. innovations **e**(*t*). We repeat the simulation 1000 times and determine how many times the zero channels report a $p < 0.05$, i.e. we numerically estimate the false discovery rate. We find a false discovery rate of approximately 0.05 for the null channel, suggesting that *p*-values are correctly estimated (Fig 4). For all others, the chance of detecting the non-zero effect is 1, i.e. a perfect true positive rate.

### 3.3 Example: Brain signals in humans

A key advantage of VARX models lies in their ability to factorize the overall system response into AR and MA components, as shown in Eq (4). This separates the influence of endogenous variables into an initial response followed by ongoing reverberations within the dynamical system. To illustrate this, we analyze intracranial electroencephalography (iEEG) recordings from a patient watching movie clips (data from [48]). We focus on 50 electrodes from visual brain regions (occipital cortex, fusiform face area, and parahippocampal cortex). We used the neural data from [48], and use the same preprocessing to extract high-frequency broadband activity (BHA, 70–150Hz, downsampled to 60Hz), often considered a marker of local neuronal firing. The exogenous "input" is a pulse train indicating fixation starts (moments of new visual input). Of course, multiple other features of the video stimulus could have been used as "input". Previous research with natural speech and movies has used features such as sound volume, visual motion, or specific content from the video and sound [49, 50]. By focusing on fixation onset, we intend to extract the neural activity associated with the initial visual processing [51].

The first observation is the diagonal terms of **A** dominate (Fig 5A), with oscillating parameters (indicating a high-pass filter) (Fig 5C). What is most evident is that the **B** (Fig 5D) response is shorter than the total system response **H** (Fig 5F). This suggests that the VARX model decomposes the total response into a fast response followed by a prolonged response due to the recurrence in the brain network. Estimating **H** directly (following [35], Fig 5E), or using the VARX model, i.e. as $\mathbf{H} = (1 - \mathbf{A})^{-1}\mathbf{B}$ (Fig 5F), we see that the two are very similar. The factorization of the total system response in the VARX model, Eq (4), thus appears to be a good approximation of the direct estimate as a purely MA system response.

We employed basis functions to represent long delay filters **B** of length $n_b$ efficiently with fewer parameters, namely $\underline{n} < n_b$ (see Section 2.7 for details).

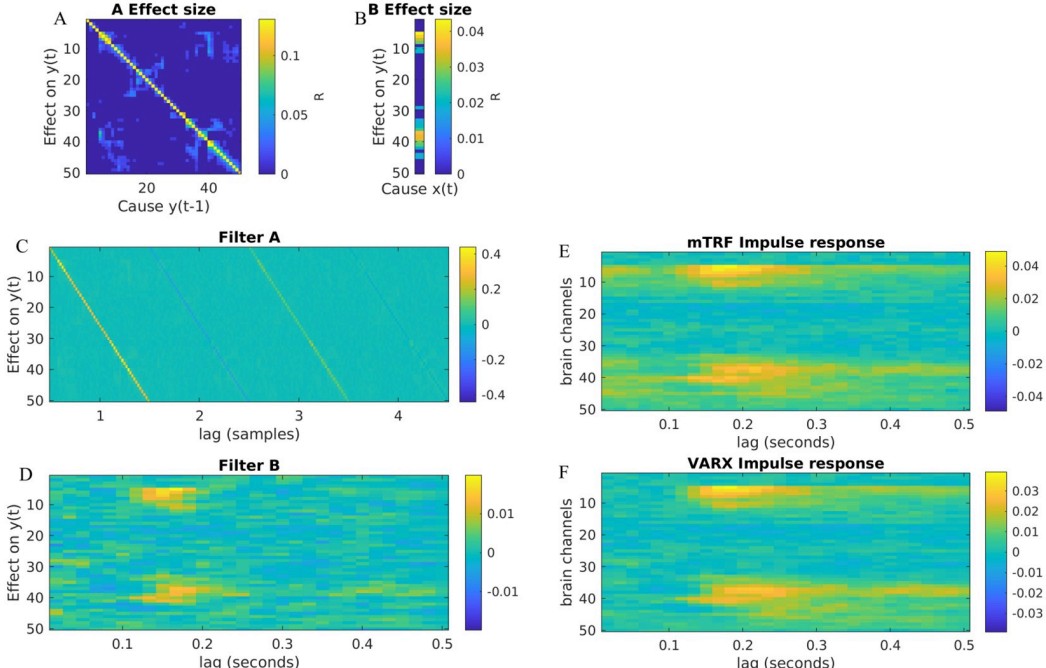

**Fig 5. Example of intracranial recording in humans: A VARX model was fitted to broadband high-frequency activity during free viewing of 7 videos recorded from 50 ($d_y$ = 50) electrodes.** A total of 43.6 minutes of data was used at a sampling rate of 60 Hz ($T$ = 156, 955) from a single patient. (A) Effect size $R$ for the recurrent connectivity **A** between recording electrodes—in the language of neuroscience, this could be called "functional connectivity". (B) Effect size $R$ of fixation onset as an exogenous variable on different electrodes. (C) **A** filter coefficients ($n_a$ = 4). (D) **B** filter coefficients $n_b = 30, \underline{n} = 20$. (E) System response estimated as a multivariate MA filter—in the language of neuroscience, this is the multivariate "temporal response function" (mTRF). (F) System response resulting from the VARX model estimate (Eq 4). Data from [48].

## 3.4 Example: Physiological signals in human

Human physiology is a dynamic system with multiple dependent signals. In previous work, we reported correlations between respiration, heart rate, pupil response, and brain activity [52]. We were motivated to identify the interactions between the body and mind in these signals. Pupil size and heart rate were measured in the experiment as metrics of physiological arousal. Respiration was measured because it is well known to affect heart rate, and eye movements were analyzed because of their association with arousal [53].

The present VARX analysis indicates potential directional effects among these physiological variables (Fig 6). As variables are added to the VARX model the connectivity structure is typically preserved. In this specific example, using a controlled breathing task, we initially observe a bidirectional link between pupil size and heart rate (Fig 6A), however, this disappears once respiration is taken into account (Fig 6B). Instead, this link is explained by an effect of respiration on pupil size, together with the well-established bidirectional link between respiration and heart rate [54] that is recovered in this data. Saccades, which are short, rapid eye movements, also have a well-established effect on pupil size [52]identified in this study (Fig 6C).

In general, adding variables can remove links—if the new common-cause variable provides an explanation for a spurious link. Adding variables can also add links—if the addition is a "collider". This is well established for i.i.d samples [55] and is no different for temporally correlated time sequence data. We will demonstrate this further using simulated data in Section 4.

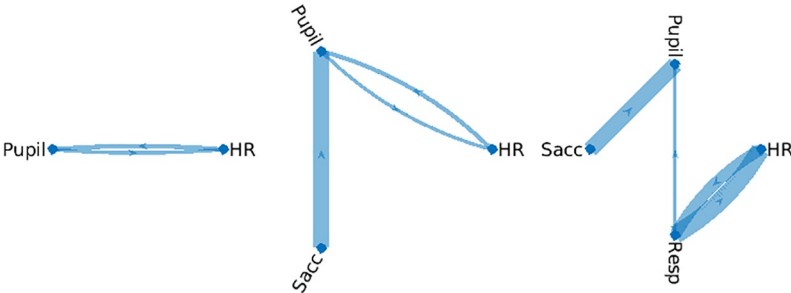

**Fig 6. Example of physiological signals in humans.** This data was collected while study participants carried out a rhythmic breading task. In this case there was no exogenous stimulus, so we only fit a VAR model. Links in **A** are shown if $p < 0.001$. Here we had 26 minutes of data compiled across multiple subject samples at 25Hz ($T = 26*60*25$). Data from [52].

## 3.5 Example: Union participation in the US

Here, we present an analysis from the field of sociology. We examine the history of workers' union membership and its relationship to strikes (Fig 7A). We hypothesize that strikes increase union membership in subsequent years. The variables here were specifically selected to test this hypothesis, largely following [56]. We assume the unemployment rate is unaffected by union variables, so it is modeled as an exogenous input. In contrast, the number of unionized workers, the number of workers on strike, and the number of strikes can all potentially influence each other. VARX Granger analysis (Fig 7B) suggests that unemployment affects unionization, which in turn affects the number of strikes, which obviously affects the number of workers on strike. These results depended on the choice of hyper-parameters $n_a$, $n_b$, $\lambda$. Only the effect of NumberOfStrikes → WorkersOnStrike was robust to parameter choice. What did not robustly emerge from this data is evidence for the initial hypothesis that strikes lead to an increase in union membership.

## 3.6 Example: Macroeconomic dynamic in the US

As a final example, we demonstrate the model on a dataset from the U.S. Federal Reserve encompassing fiscal, monetary, and labor factors, spanning quarters from 1959 to 2009. These data were selected because of their availability in the Econometrics Toolbox in MATLAB.

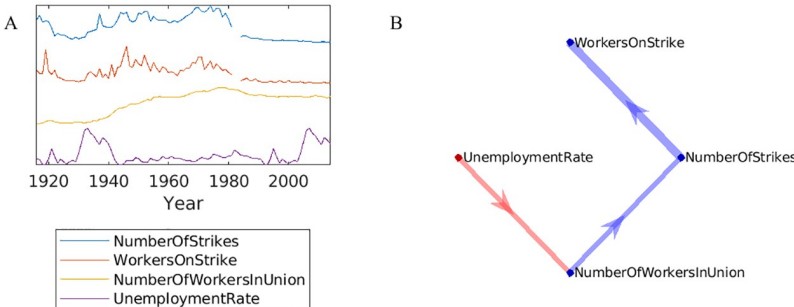

**Fig 7.** Example on union participation and strikes: (A) Historical data from the US. We treated the unemployment rate as an exogenous input in the VARX model, and the others as endogenous variables. (B) Significant effects in **A** and **B** are indicated in blue and red, respectively ($p < 0.05$, $n_a = n_b = 3$, $T = 195$). Note missing data around 1980, which was omitted during the estimation, including a 3-year history.

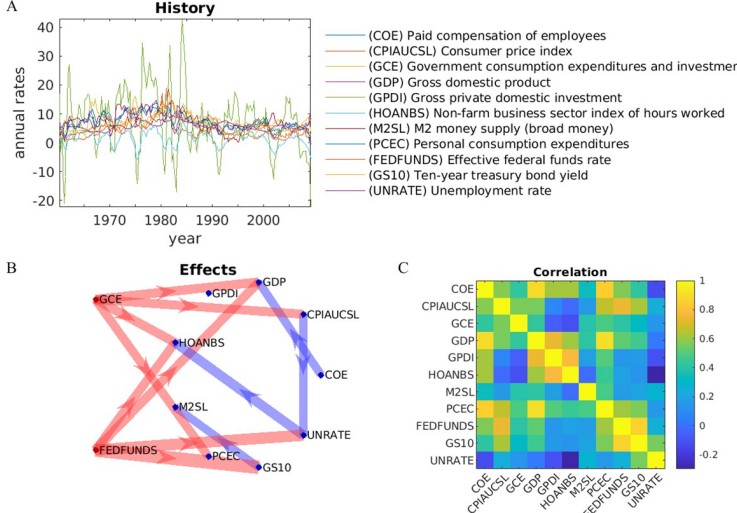

**Fig 8.** Example on US macroeconomic data: (A) Historical data from the US measured every quarter. None-rate variables (1–8) have been converted into annual percentage rates of change. (B) Significant effects in **A** and **B** are indicated in blue and red, respectively ($p < 0.001$, $T = 195$). We are taking 12–18 months history into account ($n_a = 4$, $n_b = 6$). (C) Pearson correlation of all variables.

Here we have converted all gross numbers into annual percentage rates. This removes the exponential growth resulting from predominantly positive rates (Fig 8A), which leads to trivial correlations and non-stationarity (sometimes referred to as unit-root signals). To determine the effect of the government on the economic variables, we examined the impact of government spending (GCE) and federal funds rate (FEDFUND). Government spending itself is a function of economic conditions, such as unemployment benefits, which are automatically linked to unemployment, while a rise in GDP increases tax revenue, which typically leads to increased government spending. Nevertheless, by treating GCE and FEDFUND as exogenous variables, we are asking what effects these government policies have on the economy, if they were controlled independently. Before we discuss the results (Fig 8B), it is important to note that the specific effects strongly depend on the choice of variables (gross numbers vs annual rates, endogenous vs exogenous) and parameters (independent of hyper-parameters $n_a$, $n_b$, $\lambda$). However, a robust finding is the direct effect of government spending on the gross domestic product (GDP), inflation (CPIAUCSL) and personal spending (PCEC). Rate policy affects the unemployment rate (UNRATE) independently of government spending. Despite the sensitivity to parameters, the model identifies sensible relationships and demonstrates that many variables remain independent despite a dense correlation structure (Fig 8C).

## 3.7 The case of a missing and superfluous variable

Here we want to evaluate the case where the model does not match the data generation process. We simulated three possible data generation processes with one input and two outputs ($d_x = 1$, $d_y = 2$). We simulated three cases, where the exogenous input $x$ (conditioning variable) will be either a common cause, a collider, or an independent variable (see Fig 9). In all cases, the simulation implements a one-directional effect $y_1 \rightarrow y_2$. We then measure how frequently we find $p < 0.05$ for this path, i.e. the power of the test to identify a correct path, and how frequently we find $p < 0.05$ for $y_1 \leftarrow y_2$, i.e. the rate of false discovery. Note that conditioning on a collider is known to introduce spurious correlations [55]. We test this for data generated

(A)

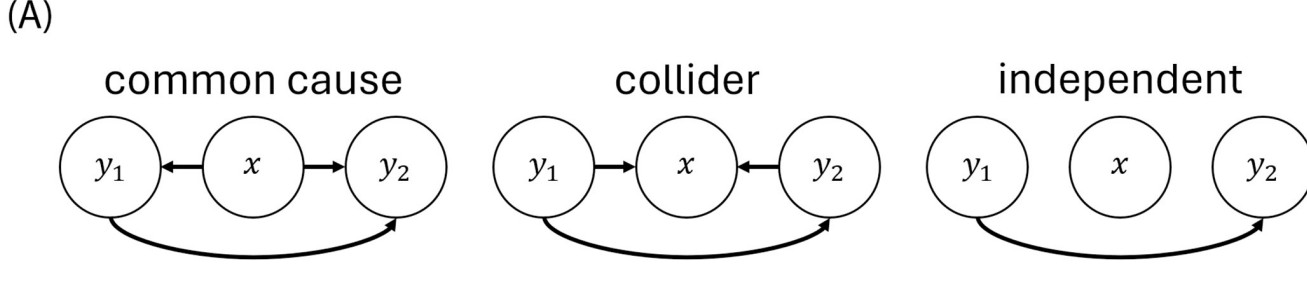

(B)

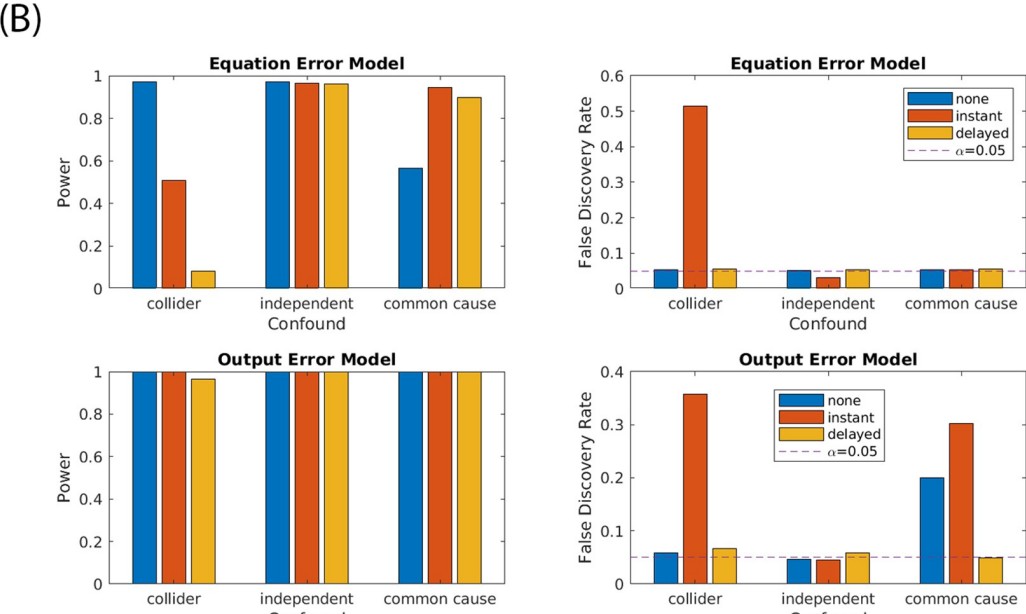

**Fig 9. The VARX (equation error) model requires a larger $T$ to obtain similar power as the output error model (22).** Simulation here used $n_a = n_b = 3$, $T = 5000$ and normal i.i.d. error.

with both the equation (VARX) and output error models. The results in Fig 9 indicate that the false discovery rate is correctly estimated at 0.05 in most scenarios, i.e. we are not finding causal effects above chance where there were none. This result holds regardless of whether $x$ was included as input (i.e. as a control variable with instant or delayed effect) or whether it did or did not have a true effect on $y_1$ and $y_2$ (common cause vs independent). Only when incorrectly modeling a collider as input, did we obtain spurious effects. Statistical power was improved when including the input to the model. In summary, there is no risk of false discovery when including input variables, even if they don't have a true effect, except if they are actually affected by the internal variables $y$.

An additional mismatch between the model and the data generation process can be the origin of the innovation process. Despite using the VARX model, when the data was generated with an output error model (Fig 9-bottom), the false discovery remains limited at the target of 0.05. However, a common unobserved cause can generate spurious effects $y_1 \leftarrow y_2$. It has been suggested that running a Granger causality model on time-reversed data provides a control for this situation [57, 58]. We have found that running the model on time-reversed data results in

spurious effects in all conditions tested here, so it is not clear to us how this can provide a remedy, and the caveat of an unobserved common cause remains when we are not directly observing endogenous variables, but only a noisy version of the internal dynamic.

## 4 Discussion

Here, we first discuss the novel contributions of this work and compare our code with existing software tools. We then follow with a number of caveats and methodological comments.

### 4.1 Novel contributions

Exogenous variables were incorporated into the Granger formalism as conditional dependence by Geweke [34] and were already briefly discussed in the original work of Granger [1]. In practice, this "conditional causality" has been used to control for spurious correlations due to common causes. The code implementations of this idea [40, 59] only use exogenous variables to remove confounds. In contrast, we propose to model the total system response as a combination of both exogenous effects and endogenous dynamics. In this view, the exogenous effects are not a nuisance, but an important component of the model to be estimated, encompassing multiple time delays. Although economists have employed VARX models to capture exogenous effects, the use of the Granger formalism to establish the effects of individual variables is not as widely used. Indeed, the correspondence of the "conditional Granger-causality" [3, 60] with VARX models is not well known. While the VARX model is common in statistics toolboxes, we are not aware of any implementation of the VARX model with the Granger-Geweke test to assess the effects.

### 4.2 Related toolboxes

Several software tools can estimate VARX models. We are not aware of one that provides the Granger-Geweke test for significance. The MVGC toolbox [40] in MATLAB supports control variables, but does not report results on exogenous variables. It therefore is mostly a tool to estimate only VAR models. The MVGC toolbox identifie the parameters **A** in both the time and frequency domain. To our knowledge, all implementations of VARX models identify parameters A and B only in the time domain, including our own. Implementations of VARX models in MATLAB Econometrics Toolbox and in SAS software for instance, make significance statements for individual delays but do not allow for the exogenous variable, i.e. $nb = 1$. Other toolboxes written in MATLAB/Python such as mTRF [35], NCRF [36], or Unfold [61] only identify parameters **B** of MA models, i.e. they ignore endogenous effects. To our knowledge, this is the first implementation of a model that estimates both **A** and **B** with time delays in each. Therefore we estimate both endogenous and exogenous delayed effects and emphasize computational efficiency to handle large datasets, with comparatively long filters **B**. The tool is available in MATLAB, Python, and R.

### 4.3 Equation error versus output error models

Estimating model parameters for a VARX model has a closed-form solution, which is much faster than finding parameters for an output error model, which requires iterative algorithms [4]. The gain in computational efficiency results from the assumption that $y(t)$ is observable. This may not be a good assumption in the case of brain signals measured across the skull, such as EEG/MEG where only a linear mixture, possibly with added noise, is observed. In that case, iterative algorithms are needed, but the Granger formalism can still be used with some effort [39]. However, in the VARX model, we do not need to assume that all internal activity is

directly observable. Any unobserved activity is captured as innovation $e(t)$. We only need to be aware that any recurrent connectivity may be due to those unobserved common "causes". In particular, symmetric effect sizes $R$ will be suggestive of such a missing variable. The role of the error is quite different in the two models. In the VARX (equation error) the error is an internal source of innovation driving the recurrent dynamic similar to the drive that comes from the input. The internal states are fully observable. In the output error model, the input entirely drives the system, and the error only affects the observations and is not injected into the dynamic.

## 4.4 Comparison of VARX, MA, OE, and VAR models

VARX and output error (OE) models can be viewed as ways to break down a system's response into moving average (MA) and autoregressive (AR) components. Alternatively, the entire system response can be modeled as a pure MA filter, as demonstrated in Fig 5E. In theory, incorporating an AR component allows for the representation of long impulse responses with fewer parameters, which is a practical advantage. However, the key difference lies in the error assumptions: MA and OE models assume errors at the output, while VARX models assume an internal innovation process with no error in the observations. Therefore, VARX models should not be considered mere input-output models, but rather models of internal dynamics. It's worth noting that all variables can be included in an AR portion of the model, allowing the estimation process to determine if any variable acts as an external input (i.e. that it does not depend on any other variable). For example, in the US macroeconomic model, arguably, government spending should have been included in the AR portion of the model, as it may depend on other variables. Including variables as exogenous serves to incorporate prior knowledge, such as knowing that movie stimuli cannot be caused by brain activity. Additionally, it allows for counterfactual analysis, such as exploring the effects of independently controlled government spending.

## 4.5 Sensitivity to parameters

A caveat to all results above is that individual links can be sensitive to the model assumptions, namely, which variables are selected as endogenous (and can be affected by all others), and which variables are selected as exogenous (and cannot be affected). An example of that was the choice of the unemployment rate as exogenous to the dynamic of unions. The results can also depend on which endogenous variables are included, as we saw in the example of physiological signals. Results can also depend on the number of parameters $n_a$ and $n_b$ and regularization factor $\lambda$ (we saw this in examples with Unions and the US macroeconomic data). Further investigation on the robustness of parameter choice is required for a clear interpretation of those results. Although we did demonstrate this here, these parameters could be established with cross-validation.

## 4.6 Caveats to causality

In Granger's original work [1], the error of the full and reduced model refers to one-dimensional signals where $y(t-1)$ is used in both cases and $x(t)$ is either used or omitted. If the error is significantly reduced by including $x(t)$ in the model, Granger argues that $x$ "causes" $y$. This interpretation is problematic for several reasons [62]. As we saw, when common causes are not observed (either as external input or internal variables) they can generate spurious links [63]. Bidirectional effects between two variables (e.g. Fig 8) may be an indication of an underlying unobserved common cause. Similarly, including colliders can cause spurious links. All this is well explained by Pearl's approach to causal inference [55]. Therefore one should not

think of the Granger formalism as serious evidence for a causal graph without a well-justified prior graphical model [64]. In particular for large dimensional datasets such as brain data, where we only observe a tiny fraction of all the variables, the risk of unobserved common causes is much too large to take the resulting graph seriously as a causal graph. Nevertheless, asymmetries in the **A** matrix can be seen as evidence of temporal precedence suggestive of an asymmetric "information flow".

## 4.7 Non-stationarity

Deviance makes a statistical judgment for the entire channel, not individual delays, as is common we simply treat each delay as a new predictor with its statistical test (this is the approach of the MATLAB VARX). There are multiple methods under the umbrella of "Granger causality" that attempt to decide on how many tabs or which delays to use. By collapsing statistical evidence into a single test statistic, Deviance, this approach has greater statistical power. This is reflected in the linearity of $D$ with the number of samples $T$. The flip side is that this statistic is very sensitive to violations of its assumptions. For instance, it assumes that all $T$ samples of the innovation process are independent and identically distributed. The AR portion of the model assures that the linear-fit residual errors are uncorrelated in time, however, if there is any non-stationarity, this will no longer be the case. Therefore, non-stationarity will cause spurious correlations [65]. In particular, any transient will cause larger deflection and correlation across samples. In particular, transient that affect several signals, say a common edge at the start or end of the signal may appear to behave like a common drive with high amplitude that results in a spurious link. Therefore, in the present approach, one has to treat edges and transients with utmost care to avoid spurious links.

Some have argued that issues with non-linearity and non-stationarity can be addressed [66]. Barnett et al. proposed a State Space model that can cope to some degree with missing variables, does not need to compute a reduced model, and can deal with non-linearity and non-stationarity [66]. However, Stokes and Purdon showed that even the state-space Granger is not immune to confounding effects, non-stationarities, etc. The topic remains a matter of debate [67].

An alternative is to avoid using analytic expressions for the $p$-value, Eq (14), and instead use standard non-parametric statistics. For time series, the simplest is to randomly time-delay channels relative to one another, potentially with a circular wraparound. All else in the model identification, i.e. estimates **A**, **B**, and effect size $R^2$ remain valid estimates of linear predictions even in the presence of non-stationarity and non-linearity.

## 4.8 L2 regularization

In this study, we proposed utilizing L2 regularization due to its compatibility with the closed-form solution of linear least squares problems. This approach enables efficient computations for large models. It is often used when estimating MA models (e.g., mTRF toolbox) as long delays add a large number of parameters. However, all regularization methods introduce a bias in the model estimates. This introduced the need for a bias correction in parametric estimates of statistical significance [68]. Corrections are available for L1 but not for L2 regularization [39, 41, 42], which we presented here for the first time. L1 regularization has the advantage that it results in sparse parameterizations, and has been used in the context of VAR models [31, 39, 41] However, it is computationally more demanding. In contrast, L2 regularization allowed us to implement fast computations of statistical significance for each channel in **A** and **B**. These new bias correction formulas should also enable fast computation of statistical

significance in MA models, which so far have not been available in existing toolboxes, e.g. mTRF or Unfold [35, 61].

## 4.9 Stability

A word about **A** is in order. The AR filter $1/(1 - \mathbf{A})$ can be unstable. We have not implemented any mechanism for this vectorial AR filter to remain stable. Lack of stability only manifests in the systems when computing the overall systems response **H**, which is not necessary during estimation for the **B**, **A** nor the calculation of statistical significance of each path (contrary to the output error model, where the recurrence has to be run back in time to estimated gradients, risking issues of stability. We rarely encountered unstable AR estimates, and where we did, L2 regularization addressed the issues. But again, there is nothing in our formalism to ensure the stability of **H**.

## 4.10 1/f spectrum

Another word about **A**. The diagonal elements of **A** in practice will always be high-pass filters, as we saw in the example of intra-cranial recordings. We advise not to take individual delays in the diagonal terms literally. The reason for this is that the innovation process is assumed to be white (constant spectrum), whereas all natural signals tend to have a 1/f spectrum. As a result, $1/(1 - \mathbf{A})$ has to have a 1/f spectrum, and **A** has to scale with f, i.e. be high-pass. In practice, we find that this is entirely accomplished by the diagonal elements of **A**. But the caveat in principle applies also to the off-diagonal elements. Future work could consider a VARMAX model where the innovation is first filtered and then injected into the recurrent dynamic [4]. However, estimation of VARMAX model parameters is a non-convex optimization problem with similar complications to the output-error model.

## 5 Conclusion

The predominant approach to modeling the effect of exogenous variables onto a dynamical system is to simply treat them as input and output of a vectorial MA filter (known as "temporal response function" in neuroscience, or simply "impulse response" in the linear systems literature). Unlike the VAR model, this is not commonly examined in the Granger formalism. Although Granger and Geweke both incorporate exogenous variables into the analysis formalism [34], the connection to the VARX model has not yet been widely recognized. We hope to have bridged this gap. While not incorrect, the simple MA approach fails to factor out the portion of the total system response that is due to the internal dynamic and separates that from the external drive. In contrast, when relying only on VAR models, one fails to exploit the prior knowledge that some variables are independent of the internal dynamic. In summary, different models vary in their assumptions about how to break down the system's overall response. When estimating with the VARX model, we manage to uniquely factor the overall response into external drive versus internal dynamics.

## Supporting information

**S1 File. Validation of p-values with L2 regularization.**
(PDF)

**S2 File. Validation of p-values when using basis functions.**
(PDF)

**S3 File. De-biased deviance under L2 regularization.**
(PDF)

## Acknowledgments

We would like to thank Jacek Dmochowski for conversations about the overall system response of the VARX model. We thank Alain de Cheveigne for providing feedback on an earlier version of this manuscript. We thank Wim Vijverberg for general advice on the analysis of macroeconomic data.

## Author Contributions

**Conceptualization:** Lucas C. Parra, Vera E. Parra, Behtash Babadi.

**Data curation:** Lucas C. Parra, Aimar Silvan, Maximilian Nentwich, Jens Madsen, Vera E. Parra.

**Formal analysis:** Lucas C. Parra, Behtash Babadi.

**Funding acquisition:** Lucas C. Parra.

**Investigation:** Lucas C. Parra.

**Methodology:** Lucas C. Parra, Vera E. Parra, Behtash Babadi.

**Project administration:** Lucas C. Parra.

**Software:** Lucas C. Parra, Aimar Silvan, Maximilian Nentwich, Jens Madsen.

**Visualization:** Lucas C. Parra, Aimar Silvan, Maximilian Nentwich, Jens Madsen.

**Writing – original draft:** Lucas C. Parra, Behtash Babadi.

**Writing – review & editing:** Lucas C. Parra, Behtash Babadi.

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
