## [Decision Letter · Decision Letter 0]

19 Aug 2024

PONE-D-24-26763VARX Granger Analysis: Modeling, Inference, and Applications

PLOS ONE

Dear Dr. Parra,

Thank you for submitting your manuscript to PLOS ONE. After careful consideration, we feel that it has merit but does not fully meet PLOS ONE’s publication criteria as it currently stands. Therefore, we invite you to submit a revised version of the manuscript that addresses the points raised during the review process.

**ACADEMIC EDITOR:**

This manuscript presents a VARX Granger analysis framework with potential applications in various fields. The authors provide the model's foundational equations, user-friendly code, and demonstrations using simulated and real-world datasets. While the manuscript shows merit, several revisions are required before it can be considered for publication in PLOS ONE.

**Key Required Revisions:**

Introduction and Literature Review: The introduction needs substantial expansion (at least 1.5-2 pages) to clearly articulate the novelty and significance of this work within the existing literature. The literature review also requires significant expansion (at least 2 pages) to comprehensively cover relevant studies and position this work within the broader context.

Methodology: The rationale for selecting specific indicators for the model needs to be strengthened with more robust theoretical explanations. The methodology section should be expanded (at least 2 pages) to provide a more detailed and clear explanation of the VARX model and its implementation.

Analysis and Discussion: The analysis section is currently too brief and simplistic. It needs to be expanded (at least 3 pages) to provide a more in-depth exploration of the results and their implications. The discussion section should be separated from the conclusion and expanded (at least 2 pages) to discuss the limitations of the study, potential future research directions, and the broader impact of the findings.

Clarity and Accessibility: The manuscript is heavily focused on mathematical details, making it challenging for a broader audience to understand. The authors need to revise the manuscript to improve clarity and accessibility for readers from various disciplines. This includes using less technical language where possible and providing more intuitive explanations of the concepts and methods.

Title: The title should be revised to be more specific and include the application areas explored in the manuscript.

Abstract: The abstract should be condensed and focus on the key findings and contributions of the study.

Addressing Reviewer Comments:

Reviewer 1: The concerns raised by Reviewer 1 regarding the limited scope of the literature review, methodology, analysis, and discussion align with the major required revisions outlined above. The authors should address these concerns by expanding these sections and providing more detailed explanations.

Reviewer 2: The authors should address the specific points raised by Reviewer 2 regarding:

AR and MA components: Clarify how these components in the VARX model relate to mTRF analysis and frequency analysis in neuroscience.

mTRF toolboxes: Discuss various mTRF toolboxes in comparison to the proposed VARX model.

L2 Regularization: Contextualize the discussion on L2 regularization within the framework of regularization parameters in mTRF analysis.

Data Availability:

Reviewer 1: The authors need to address the data availability concerns raised by Reviewer 1 and ensure that all data underlying the findings are made fully available as per PLOS ONE's data policy.

Overall:

The manuscript presents a potentially valuable contribution to the field. However, substantial revisions are required to address the concerns raised by the reviewers and meet the publication standards of PLOS ONE. The authors are encouraged to carefully consider the feedback provided and revise the manuscript accordingly.

We look forward to receiving your revised manuscript.

Kind regards,

Abdullah Mohammad Ghazi Al khatib, Ph.D.

Academic Editor

PLOS ONE

 [LCP, ASO, JM, National Institutes of Health, NIMH P50 MH109429

LCP, JM, National Science Foundation, DRL-2201835].  

Reviewers' comments:

Reviewer's Responses to Questions

**Comments to the Author**

1. Is the manuscript technically sound, and do the data support the conclusions?

Reviewer #1: Partly

Reviewer #2: Yes

2. Has the statistical analysis been performed appropriately and rigorously? 

Reviewer #1: Yes

Reviewer #2: Yes

3. Have the authors made all data underlying the findings in their manuscript fully available?

Reviewer #1: No

Reviewer #2: Yes

4. Is the manuscript presented in an intelligible fashion and written in standard English?

Reviewer #1: No

Reviewer #2: Yes

5. Review Comments to the Author

Reviewer #1: Journal: PLOS ONE

Article title: VARX Granger Analysis: Modeling, Inference, and Applications

Manuscript ID: PONE-D-24-26763

General Comments:

This article studies the disentangling delayed external effects from internal dynamics. The authors use a Vector Autoregressive model with eXogenous input (VARX) by providing fundamental equations, user-friendly code, and demonstrations using simulated and real-world data from neuroscience, physiology, sociology, and economics. The authors reached the conclusions that model avoids spurious correlation by factoring out external influences from internal dynamics, leading to more parsimonious explanations of the systems with methods for enhancing model efficiency, such as L2 regularization for limited data and basis functions to cope with extended delays

Overview:

The paper is good written and the empirical work does appear to be carefully and correctly done. The research question is somehow good and it does make a sufficient new contribution to the literature to be suitable for the PLOS ONE ONLY after MINOR revisons.

In fact, the use of VARX Granger Analysis: Modeling, Inference, and Applications is quite new in the literature.

The contribution of the paper is the use of a Vector Autoregressive model with eXogenous input (VARX) by providing fundamental equations, user-friendly code, and demonstrations using simulated and real-world data from neuroscience, physiology, sociology, and economics.

The paper is neutral interesting; and in my view, it needs to be MINOR improved to reach the standard required for publication in this journal.

Specific Comments:

1. Title: introduce the subjects for application

2. Abstract: somehow large, try to reduce and introduce with the present results from the article

3. Introduction: NOVELTY + results (better explanation); enlarge at least 1.5-2 pages

4. Literature review: at least 2 pages

5. Methodology: why the authors use only these indicators into the model? Present some solid theoretical explanations for these indicators. Enlarge at least 2 pages.

6. Section 4: The analysis is very small and simplistic; enlarge this part of the article to at least 3 pages

7. Discussions: at least 2 pages; separate from the conclusions

8. The article is extremely detailed and concentrated on results part; modify and transform into an academic one

General considerations:

The idea of the article is good, and the construction of the article is sometimes very mathematical and not easy to read. The authors MUST improve the literature, methodology, explanations, discussions, and change the article accordingly. The authors MUST rearrange the methodology part, enlarge the analysis, modify into a readable article.

I ONLY recommend this article be published in PLOS ONE after MINOR revisions (literature, methodology, analysis and the discussion).

Reviewer #2: This study presents a novel approach for capturing temporal dynamics within a dynamic system. Specifically, the VARX model decomposes the system response into auto-regressive (AR) and moving average (MA) components, offering potential utility in the field of neuroscience.

While the authors provide detailed equations for the VARX model, the exposition on separating the AR and MA components is lacking in specificity. It is recommended that the authors elaborate on how the AR and MA components in the VARX model relate to mTRF analysis and frequency analysis in the neuroscience domain.

Furthermore, a discussion on various mTRF toolboxes in comparison to the VARX model is warranted.

Lastly, the discussion on L2 regularization should be contextualized within the framework of regularization parameters in mTRF analysis.

6. PLOS authors have the option to publish the peer review history of their article (what does this mean?). If published, this will include your full peer review and any attached files.

Reviewer #1: No

Reviewer #2: **Yes: **Ling Liu

---

## [Author Response · Author response to Decision Letter 0]

18 Sep 2024

see attached detailed response document

---

## [Decision Letter · Decision Letter 1]

4 Nov 2024

VARX Granger analysis: Models for neuroscience,

physiology, sociology and econometrics

PONE-D-24-26763R1

Dear Dr. Parra,

We’re pleased to inform you that your manuscript has been judged scientifically suitable for publication and will be formally accepted for publication once it meets all outstanding technical requirements.

Kind regards,

Abdullah Mohammad Ghazi Al khatib, Ph.D.

Academic Editor

PLOS ONE

Reviewers' comments:

Reviewer's Responses to Questions

**Comments to the Author**

1. If the authors have adequately addressed your comments raised in a previous round of review and you feel that this manuscript is now acceptable for publication, you may indicate that here to bypass the “Comments to the Author” section, enter your conflict of interest statement in the “Confidential to Editor” section, and submit your "Accept" recommendation.

Reviewer #3: All comments have been addressed

2. Is the manuscript technically sound, and do the data support the conclusions?

Reviewer #3: Yes

3. Has the statistical analysis been performed appropriately and rigorously? 

Reviewer #3: Yes

4. Have the authors made all data underlying the findings in their manuscript fully available?

Reviewer #3: Yes

5. Is the manuscript presented in an intelligible fashion and written in standard English?

Reviewer #3: Yes

6. Review Comments to the Author

Reviewer #3: The reviewed article complies with the standards imposed by the journal PLOS ONE.

The language, structure, accuracy of the demonstrations, data and analyzes presented facilitate the reading and easy understanding of the content of the article, as far as possible, by the general public.

In conclusion, we recommend its publication in the current format.

7. PLOS authors have the option to publish the peer review history of their article (what does this mean?). If published, this will include your full peer review and any attached files.

Reviewer #3: No

---

## [Editor Report · Acceptance letter]

22 Dec 2024

PONE-D-24-26763R1 

PLOS ONE

Dear Dr. Parra, 

I'm pleased to inform you that your manuscript has been deemed suitable for publication in PLOS ONE. Congratulations! Your manuscript is now being handed over to our production team.

Kind regards, 

on behalf of

Dr. Abdullah Mohammad Ghazi Al khatib 

Academic Editor

PLOS ONE